# RELEVANCE-BASED EMBEDDINGS
# FOR EFFICIENT RELEVANCE RETRIEVAL

## ABSTRACT

In many machine learning applications, the most relevant items for a particular query should be efficiently extracted. The relevance function is typically an expensive neural similarity model making the exhaustive search infeasible. A typical solution to this problem is to train another model that separately embeds queries and items to a vector space, where similarity is defined via the dot product or cosine similarity. This allows one to search the most relevant objects through fast approximate nearest neighbors search at the cost of some reduction in quality. To compensate for this reduction, the found candidates are then re-ranked by the expensive similarity model. In this paper, we propose an alternative approach that utilizes the relevances of the expensive model to make *relevance-based embeddings*. We show both theoretically and empirically that describing each query by its relevance for a set of support items creates a powerful query representation. We also investigate several strategies for selecting these support items and demonstrate that additional significant improvements can be obtained. Our experiments on diverse datasets show improved performance over existing approaches.

## 1 INTRODUCTION

Finding the most relevant element (item) $i$ to a query $q$ among a large set of candidates $I$ is a key task for a wide range of machine learning problems, for example, information retrieval, recommender systems, question-answering systems, or search engines. In such problems, the final score (relevance) is often predicted by the pairwise function $R : Q \times I \to \mathbb{R}$, where $Q$ is a query space and $R$ approximates some ground truth relevances from training data, such as click probability, time spent or else. Depending on the task, $Q$ can be, for example, a set of text queries or a set of attributes — numerical features describing the user, such as age, time spent on the service, etc. An item $i \in I$ could also be represented in various ways, including feature vectors.

The problem of relevance retrieval for a query $q$ can be written as $\arg\max_{i \in I} R(i, q)$. For practical applications, it is usually required to return not one but $K$ best items (for directly displaying to the user or further re-ranking), which could be written as

$$\text{Best}_K(R, q) := \underset{T_I \subset I, |T_I| = K}{\arg\max} \left( \underset{i \in T_I}{\arg\min} R(i, q) \right).$$

Most recommendation systems are characterized by a large size of the item space $I$ (millions to hundreds of millions), so an exhaustive search is not feasible. This problem is often solved by training an auxiliary model $\tilde{R}$, called a Siamese, two-tower, or dual encoder, in which late binding is used: $\tilde{R}(i, q) = S(F_I(i), F_Q(q))$, where $F_I : I \to \mathbb{R}^d$, $F_Q : Q \to \mathbb{R}^d$, and $S$ is some lightweight similarity measure, usually dot product or cosine similarity.

While a lot of effort has been put into developing dual-encoder models, the cross-encoder ones are generally more powerful (Wu et al., 2019; Yadav et al., 2022). Moreover, in practice, one may have *pairwise* features that describe a query-item pair. For instance, in information retrieval, pairwise features can include statistics based on counts of each query term in the document. Clearly, such features cannot be used by dual encoders.

In this paper, we propose a solution to the problem discussed above. The main idea is to build embeddings for queries based on their relevance to some pre-selected support (or key/anchor) items

and vice versa. We theoretically show that such relevance-based embeddings are expressive enough to approximate any continuous relevance function. Importantly, our approach does not change the general pipeline of embedding queries and items into a certain space and then searching for the nearest vectors in this space via any efficient nearest neighbor search algorithm. Additionally, given relevance vectors, embeddings can be obtained via simple encoders such as MLPs (multilayer perceptrons), thus requiring little development efforts compared to standard dual encoder training with a complex set of features, for example, in the case of recommendation systems (Covington et al., 2016).

An important aspect of our approach is how to properly choose support elements. We investigate different options and show that better choices allow one to significantly boost the overall performance. In particular, even very simple strategies like clustering the elements and choosing the cluster centers as support items already give significant improvements. The results can be further improved if the elements are greedily chosen to optimize the accuracy of relevance approximation.

To evaluate the performance of the proposed embeddings, we conduct experiments on textual and recommendation datasets. We compare our approach with dual encoders and with a recent approach based on the query-item relevance matrix factorization (Yadav et al., 2022). We get an average improvement of 33% over this baseline for various data sets (from 8% to 69% see Table 3).

## 2 RELATED WORK

In this section, we discuss research areas and representative papers related to our study.

**Relevance retrieval problem** is widespread in the context of building information retrieval systems (Kowalski, 2007), such as text search engines (Huang et al., 2013), image search (Gordo et al., 2016), entertainment recommendation systems (Covington et al., 2016), question answering systems (Karpukhin et al., 2020), e-commerce systems (Yu et al., 2018), and other practical applications.

Usually, such problems are solved by learning **queries and items embeddings** into a certain space and then searching for approximate nearest elements in this space, followed by rearrangement using a heavier ranker. In particular, the works mentioned above (Covington et al., 2016; Huang et al., 2013) explicitly use this approach, offering two-tower models (a.k.a. dual encoders). Note that there are simple alternatives to the dual encoder that use, e.g., BM25 scores applicable to texts (Logeswaran et al., 2019; Zhang & Stratos, 2021) or other cheaper or more expensive alternatives (Humeau et al., 2019; Luan et al., 2021). However, there is usually trading-off complexity for quality. It is also worth mentioning the works trying to facilitate the training of the dual encoder through distilling a heavier ranker model (Wu et al., 2019; Hofstätter et al., 2020; Lu et al., 2020; Qu et al., 2020; Liu et al., 2021).

As for the **nearest neighbors search** in a common query-item space, a wide variety of algorithms exist, including locality-sensitive hashing (LSH) (Indyk & Motwani, 1998; Andoni & Indyk, 2008), partition trees (Bentley, 1975; Dasgupta & Freund, 2008; Dasgupta & Sinha, 2013), and similarity graphs (Navarro, 2002). LSH-based and tree-based methods provide strong theoretical guarantees, however, it has been shown that graph-based methods usually perform better (Malkov & Yashunin, 2018), which explains their widespread use in practical applications.

Another research direction is methods that combine nearest neighbors search with heavy ranker calls (Morozov & Babenko, 2019; Chen et al., 2022) instead of separately embedding queries and items in a common space where the search for the nearest items can be efficiently performed. Such methods show better quality in comparison with separate embeddings, however, their practical application may be limited due to a significant change in the structure of the search index. In particular, in practice, microservices with neural networks and microservices with document indexes are different services, which allows for increasing GPU utilization on the one hand and using specialized (including sharded) solutions with a large amount of memory on the other. Therefore, in this paper, we focus on the basic scenario with a separate investment in space and a separate search for the nearest elements in it.

The paper by Yadav et al. (2022) is the most relevant for our research. The idea is to apply the matrix factorization to the query-item relevance matrix in order to represent it as a product of its submatrix

containing only a few columns (relevances for random support items set) and some other, explicitly computable. Despite the simplicity of the idea and implementation, the authors have shown in detail the superiority of their algorithm over more complex approaches, such as dual encoders. We use their work (Yadav et al., 2022) as a baseline in our experiments. Note that Morozov & Babenko (2019) also propose an idea with the allocation of support elements randomly. Combining their algorithm with our strategies of selecting support items is an option for further research. As an enhancement of the original approach, in another paper, Yadav et al. (2023) proposed the idea of selecting support items per each query independently, but the time complexity of each query processing becomes linear in the number of elements, which makes the approach infeasible in most practical applications. On the contrary, our support items selection is performed in the pre-processing stage and does not increase the query time.

## 3 RELEVANCE-BASED EMBEDDINGS

In general, the information (attributes) used to calculate the ground-truth item-to-query relevances $R(i, q)$ can be divided into three types: depending only on the query $q$, only on the item $i$, and depending on both of them. The key problem when constructing separate embeddings of items and queries in the common space (that can be used for searching for the nearest elements) is the inability to use information that depends on both query and item, which lowers the quality of relevance search. Below we propose a theorem stating that, under certain constraints, it is possible to design new factors based on the query relevances to a pre-selected set of items and, vise-versa, item relevances based on a pre-selected set of queries, such that the relevance function can be well estimated using only such individual vectors. Motivated by that, we propose using relevance-based embeddings (RBE) and discuss how pre-selected sets of queries and items can be chosen in practice.

### 3.1 UNIVERSALITY OF RELEVANCE-BASED EMBEDDINGS

In this section, we prove a formal statement that any continuous relevance function of queries and items can be well approximated by a neural architecture that uses only individual relevance vectors of queries and items.

Formally, let $Q$ and $I$ be compact topological spaces of queries and items, respectively. Assume that we are given a relevance function $R : I \times Q \to \mathbb{R}$. In practice, $R$ is our relevance model which may be heavy or rely on pairwise features.

Let $S_I \subset I$ and $S_Q \subset Q$ be some finite ordered sets of *support items* and *support queries*:

$$S_I = \{i_1, \ldots, i_m\}, \quad S_Q = \{q_1, \ldots, q_n\}.$$

Let $R(i, S_Q)$ be a *relevance vector* of the item $i$ w.r.t. the set of support queries $S_Q$:

$$R(i, S_Q) = (R(i, q_1), \ldots, R(i, q_n)).$$

Similarly, $R(S_I, q)$ is a relevance vector of the query $q$ w.r.t. the set of support items $S_I$:

$$R(S_I, q) = (R(i_1, q), \ldots, R(i_m, q)).$$

Additionally $R(S_I, Q)$, $R(I, S_Q)$ - are relevance matrices composed in a similar way.

We say that a function on $I$ is a *relevance-based embedding* if it has a representation of the form $e_I(i) = f_I(R(i, S_Q), \theta_I)$ where $S_Q$ is a set of support queries and $f_I$ is some ML-architecture with parameters $\theta_I$ which parametrizes mapping $\mathbb{R}^n \to \mathbb{R}^d$. Analogously, $e_Q(q) = f_Q(R(S_I, q), \theta_Q)$ is a relevance-based embedding of the query $q$.

The following theorem holds (the proof can be found in Appendix A.1).

**Theorem 1.** *Let $I$ and $Q$ be compact topological spaces, and $R : I \times Q \to \mathbb{R}$ be a continuous function. Then, $R$ can be uniformly approximated up to an arbitrarily small absolute error by a function $\tilde{R}(i, q)$:*

$$\tilde{R}(i, q) = \langle\, f_I(R(i, S_Q), \theta_I),\ f_Q(R(S_I, q), \theta_Q)\,\rangle, \tag{1}$$

*where $S_I \in I$ and $S_Q \in Q$ are some finite sets of support items and queries and $f_I$, $f_Q$ are neural architectures with the universal approximation property (e.g., MLPs).*

$$R(I,Q)^T \qquad\qquad R(I,S_Q) \qquad\qquad R(S_I,Q)^T$$

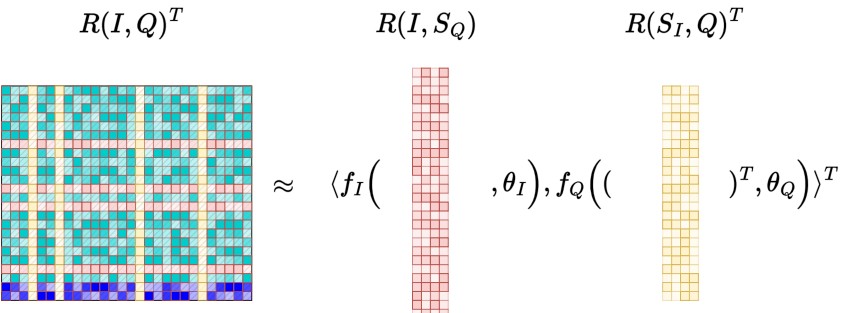

Figure 1: Relevance-based framework visualization: support queries are red, support items are yellow, and the remaining cells of the relevance matrix can be approximated by the dot product of transformed relevance vectors. The test queries are colored blue, their relevance scores for the support items are needed to approximate the remaining values.

The statement of the theorem is visualized in Figure 1. This theorem shows that the true relevance function can be approximated with arbitrary precision by some functions of the relevance vectors. To illustrate the importance of this observation, recall that usually the relevance function $R(\cdot, \cdot)$ is a complex cross-encoder model. Moreover, such a model can use pairwise features that are available only for query-item pairs. In contrast, dual encoders are known to often be less powerful and they do not have access to pairwise features. However, Theorem 1 shows that if we replace the original features with the relevance-based vectors, a dual-encoder model can well approximate the cross-encoder function.

The corollary below shows that the retrieval of the most $R$-relevant items with tolerance to $\varepsilon$-sized relevance loss can be reduced to the standard nearest neighbor search *on a sphere*.

**Corollary 1.** *For each $\varepsilon > 0$, there is a multiplier $a \in \mathbb{R}$ such that $a\tilde{R}$ is an $\varepsilon$-approximation of $R$ and $\tilde{R}$ uses embeddings scaled to the unit sphere.*

We refer to Appendix A.2 for the proof.

## 3.2 SELECTING SUPPORT ITEMS

Let us revise the statement of Theorem 1. The theorem states that there exist such sets $S_Q$, $S_I$ that the relevance function $R$ could be effectively approximated by separated embedders $F_I$, $F_Q$. In the related works (Morozov & Babenko, 2019; Yadav et al., 2022), this selection is random, implicitly assuming that the elements are in some sense equivalent. However, for example, when building a recommendation service, the popularity of different objects has a strongly skewed distribution, which is why more information is known about a small set of highly popular items than about a large set of unpopular ones. Thus, it is natural to assume that the choice of support items may have a significant effect on performance. We investigate this direction and compare the following options:

- **Random:** Support items are chosen uniformly at random (Morozov & Babenko, 2019; Yadav et al., 2022). For better reproducibility, we also present the results of using the first $|S_I|$ elements as support elements, assuming that the order of queries is pseudorandom in nature.

- **Popular:** As mentioned above, a recommendation service usually has a small set of very popular elements that many users interact with. As a result, a lot of information can be collected from these interactions thus making the popular elements more informative. Since it is not always possible to get popularity explicitly, we consider the following surrogate: choose the objects with the highest average relevance for the training set.

- **Clusters centers:** When it comes to the allocation of a representative subset of vectors, it is reasonable to consider the allocation of clusters. We consider various clustering algorithms and select the cluster centers as support elements. The number of clusters is set to the number of required support elements.

- **Most diverse**: This strategy is a greedy algorithm optimizing the minimum distance between the support elements. We first choose the element furthest from the center (by Euclidean distance) and then, at each step, an element is selected whose minimum distance to the current support elements is maximal.

- $l_2$-**greedy**: A natural step further when selecting the key elements is to optimize them so that we better approximate the relevance matrix $R(i, q)$. In this strategy, we greedily select items so that the MSE error of the CUR-approximation (Mahoney & Drineas, 2009) is minimized for the train queries. We refer to Appedinx B for the details.

### 3.3 RELEVANCE-BASED EMBEDDINGS IN PRACTICE

In this section, we discuss several aspects related to applying relevance-based embeddings in practical applications.

First, we note that relevance-based embeddings can naturally handle scenarios where the set of items changes frequently. The embeddings $f_I(R(i, S_Q), \theta_I)$ can be easily calculated for the new items without the need of re-training the embedding model $f_I$ (similarly to feature-based dual encoders).

However, in many practical tasks of information retrieval, the set of initial objects $I$ is finite: e.g., the set of movies currently available in a recommender service. In this case, the transformation $f_I(R(i, S_Q), \theta_I)$ can be replaced with trainable embeddings $\theta_I(i) \in R^d$. In contrast to items, the query set $Q$ cannot be assumed to be finite: e.g., queries can be represented by texts of unlimited length or be characterized by real-valued features.

Note that in practice, the mapping $f_Q$ (or $f_I$) could be extended by enriching it with the features of the original query (or item), which can further improve the approximation of the ground truth relevance. In other words, although the theorem states that the relevance of supporting elements is sufficient, in practice we can add more information, for example, in order to reduce the size of the support items set $S_I$ while maintaining quality.

Since the heavy ranker $R$ is assumed to be the most expensive part in terms of computational complexity, during the calculation of $\tilde{R}$, we are mainly limited by the sizes of the support sets $S_Q$ and $S_I$. In contrast, calculating $f_Q$, $f_I$, or the dot product between them is assumed to be significantly cheaper. Thus, $f_Q$ and $f_I$ may embed the relevance vectors in a higher-dimensional space, which may help to eliminate the disadvantages of the dot product, in comparison with other (Shevkunov & Prokhorenkova, 2021) ways of measuring the distances or similarities between objects in spaces.

**Scalability** Let us discuss the scalability of the proposed approach. The first step of our procedure is selecting support items. We note that this should be done once at the preprocessing stage and time restrictions are usually not strict. However, let us note that $l_2$-greedy requires computing the relevance scores for some train queries and all items. If this is infeasible, one can use downsampling to reduce the number of candidates for support items. Our preliminary experiments show that even significant downsampling gives reasonable performance of the obtained support items. If downsampling is not applicable, the proposed clustering-based approaches can be easily scaled by using external information instead of relevance vectors: in most services, items are annotated with categories that can be used as clusters or are described by feature vectors that can be used for clusterization.

Given the relevance vectors, the training does not significantly differ from any dual-encoder-like models that are commonly used in production recommendation services. The only major difference is that relevances to fixed support items should be provided.

At the inference, we need to compute the query representation, which requires $d$ relevance computations. Then, the inference is similar to dual encoders: the item representations are pre-calculated and placed in the Approximate Nearest Neighbours index like HNSW, which accepts the embedding of the query as input. These $d$ additional computations are taken into account in our experiments: we show that the proposed approach is still more efficient than standard dual encoders.

Table 1: Dataset sizes

|  | **Yugioh** | **P.Wrest.** | **StarTrek** | **Dr.Who** | **Military** | **RecSys** | **RecSysLT** |
|---|---|---|---|---|---|---|---|
| items | 10031 | 10133 | 34430 | 40281 | 104520 | 16514 | 16514 |
| queries (used) | 3374 | 1392 | 4227 | 4000 | 2400 | 6958 | 6958 |

## 4 EXPERIMENTS

In this section, we evaluate the performance of the proposed relevance-based embeddings and compare our approach with existing methods.[1]

### 4.1 EXPERIMENTAL SETUP

In our experiments, we use two groups of data: ZESHEL zero-shot entity linking datasets and production data from a recommendation service.[2] In both datasets, there is a heavy ranker that provides relevance, which we consider close to the ground truth, according to which a complete table ($R : |I| \times |Q| \rightarrow \mathbb{R}$) of relevance scores is built. In both cases, the task is to find the most relevant items for some query. The quality is evaluated as

$$HitRate(P, T) := \frac{1}{|Q_{test}|} \sum_{q_i \in Q_{test}} \frac{|Best_P(\hat{R}, q_i) \cap Best_T(R, q_i)|}{|Best_T(R, q_i)|},$$

$$HitRate(K) := HitRate(K, K),$$

where $Best_K(\mathcal{R}, q) \subset I$ is defined as the set of $K$ items $i_1, \ldots, i_K$ with the highest relevances $\mathcal{R}(i_1, q), \ldots, \mathcal{R}(i_K, q)$ to a given query $q \in Q$; $Q_{test}$ is a set of test queries that do not participate in the training: $S_Q \subset Q_{train}$, $Q_{train} \bigsqcup Q_{test} = Q$. For all our experiments, $|Q_{test}| \approx 0.3|Q|$, $|S_I| = 100$.

#### 4.1.1 ZESHEL DATASET

The Zero-Shot Entity Linking (ZESHEL) dataset was constructed by Logeswaran et al. (2019) from Wikia. The task of zero-shot entity linking is to link mentions of objects in the text to an object from the list of entities with related descriptions. The dataset consists of 16 different domains. Each domain contains disjoint sets of entities, and during testing, mentions should be associated with invisible entities solely based on entity descriptions. We run experiments on five domains from ZESHEL selected by Yadav et al. (2022). As a heavy ranker $R$, we use the cross-encoder trained by Yadav et al. (2022) and publicly available. Table 1 shows the dataset statics for the domains used in this paper.

#### 4.1.2 RECSYS DATA

To evaluate the generalization of the proposed approach to other tasks and domains, we collected a dataset from a production service providing recommendations of items to users. As a heavy ground-truth ranker $R$, we use the CatBoost gradient boosting model (Prokhorenkova et al., 2018) trained on a wide range of features, including categories and other static attributes of items, social information (age, language, etc.) of users, simple item statistics, user statistics, real-time statistics on user and element interaction, factors derived from the matrix factorizations and multiple two-tower neural networks, receiving the factors listed above as their factors.

Two versions of the dataset are presented. In the first one, CatBoost was trained to predict the time that a user is going to spend on a given item immediately after the click (in one session). In the second version, CatBoost was trained on the pairwise PairLogit target to predict the item with the

---

[1]The code and experimental data will be made publicly available after the blind review due to anonymity considerations.

[2]Not specified to preserve anonymity.

longest time spent for some long time after the click (including new sessions). The first version is denoted in the tables as RecSysLT and the second as RecSys.

This dataset allows us to evaluate the generalizability of our approach across different domains and different types of heavy rankers since gradient boosting models differ significantly from neural approaches.

### 4.1.3 BASELINE

As our main baseline, we consider the AnnCUR recommendation algorithm (Yadav et al., 2022): in this approach, the approximated relevances for a query $q$ can be re-written in our notation as:

$$\tilde{R}(I, q) := \langle R(I, Q_{train}) \times \text{pinv}(R(S_I, Q_{train})), R(S_I, q) \rangle,$$

where $\text{pinv}(X)$ is the pseudo-inverse matrix of $X$, $Q_{train}$ is a subset of the query set $Q$. Note that this formula fits our framework (1) with

$$S_Q := Q_{train}, \theta_I := \text{pinv}(R(S_I, Q_{train})),$$

$$f_I(R(i, S_Q), \theta_I) := R(i, S_Q) \times \theta_I, f_Q(R(S_I, q), \theta_Q) := R(S_I, q).$$

Thus, the predicted relevances obtained by the AnnCUR algorithm are a special case of relevance-based embeddings with relatively simple transformation functions based on matrix factorization.

What is important for further discussion, a broad comparison of this method with different basic approaches, including various dual encoders, is carried out by Yadav et al. (2022). In most of our experiments, we rely on these results, comparing only with AnnCUR. However, we explicitly provide the comparison with our best dual encoder for the new data in Section 4.4.

Note that our results generally reproduce the pipeline of the experiments in Yadav et al. (2022), in particular, the recalculated (using our preprocessing code) AnnCUR metrics are comparable with the metrics from the original paper.[3] This allows us to assert that the approaches are compared under the closest possible conditions.

### 4.1.4 RELEVANCE-BASED EMBEDDINGS SETUP

In this section, we discuss our implementation of the relevance-based embeddings.

As a trainable mapping $f_Q(R(S_I, q), \theta_Q)$, we use the following variant:

$$f_Q(R(S_I, q), \theta_Q) := R(S_I, q) \,||\, F_Q^{mlp}(R(S_I, q), \theta_Q) \,||\, (1),$$

where $F_Q^{mlp}$ is a 2-layer perceptron with the ELU activations, $||$ is the vector concatenation, and the last term is needed to represent the items offsets as a scalar product. The intuition here is that we split the representation into the prediction of AnnCUR and the trainable prediction of its error. In the experiments, such decomposition improves the convergence and training stability.

For the item mapping $f_I(R(i, S_Q), \theta_I)$, we use the following function:

$$f_I(R(i, S_Q), \theta_I) := t_I(R(i, S_Q), \theta_I) \,||\, F_I^{mlp}(t_I(R(i, S_Q), \tilde{\theta}_I)) \,||\, (c_i),$$

$$t_I(R(i, S_Q), \theta_I) := R(i, S_Q) \times P, \ P = \text{pinv}(R(S_I, Q_{train})), \ \theta_I := (P, c, \tilde{\theta}_I),$$

where $c$ is a trainable bias vector, $\tilde{\theta}_I$ — perceptron trainble parameters. Although, as noted in Section 3.3, the transformation $f_I$ acts in practice on a finite set $I$ of elements and can be learned as an embedding matrix, the approach described above greatly accelerates the speed and stability of learning.

The mappings are trained using the Adam algorithm to optimize the following loss function inside the batches:

$$L := \frac{1}{|Q_{train}|} \sum_{q \in Q_{train}} \text{softmax}(\tilde{R}(q, I))(2 \cdot \mathbf{1}_{\text{binRelevance(q)}} - 1),$$

---

[3]Our $HitRate(k, kr)$ is equivalent to Top-k-Recall@$kr$ from Yadav et al. (2022).

Table 2: Support item selection applied to AnnCUR, HitRate(100) (greater is better) is reported

| Selection strategy | Yugioh | P.Wrest. | StarTrek | Dr.Who | Military | RecSys | RecSysLT |
|---|---|---|---|---|---|---|---|
| random (AnnCUR) | 0.4724 | 0.4280 | 0.2287 | 0.1919 | 0.2455 | 0.6697 | 0.5842 |
| first 100 | 0.4845 | 0.4182 | 0.2489 | 0.1975 | 0.2599 | 0.6490 | 0.5609 |
| popular | 0.2429 | 0.3001 | 0.1154 | 0.1197 | 0.1907 | **0.7623** | **0.6695** |
| KMeans | 0.5083 | **0.4850** | 0.3226 | **0.2517** | **0.3042** | 0.7070 | 0.6184 |
| BisectingKMeans | 0.4825 | 0.4592 | 0.2839 | 0.2159 | 0.2752 | 0.7035 | 0.6213 |
| MiniBatchKMeans | 0.5077 | 0.4737 | 0.2912 | 0.2365 | **0.2826** | 0.7033 | 0.5981 |
| AgglomerativeClustering | **0.5105** | **0.4911** | **0.3264** | **0.2531** | 0.2448 | 0.7050 | **0.6265** |
| SpectralCoclustering | 0.4618 | 0.4443 | 0.2540 | 0.2076 | 0.2551 | 0.6998 | 0.6094 |
| SpectralBiclustering | 0.4654 | 0.4708 | 0.2628 | 0.1845 | 0.2533 | **0.7409** | 0.5972 |
| SpectralClusteringNN | 0.5087 | 0.4690 | 0.2742 | 0.2048 | 0.2507 | 0.6936 | 0.5740 |
| ByMin | **0.5333** | 0.4290 | **0.3325** | 0.2278 | 0.2483 | 0.6504 | 0.6182 |
| Greedy | **0.5618** | **0.5119** | **0.3677** | **0.2960** | **0.3357** | 0.7197 | **0.6565** |

Table 3: Evaluating neural relevance-based embeddings, HitRate(100) (greater is better) is reported

| Model | Yugioh | P.Wrest. | StarTrek | Dr.Who | Military | RecSys | RecSysLT |
|---|---|---|---|---|---|---|---|
| Popular | 0.0917 | 0.2410 | 0.0884 | 0.0821 | 0.1127 | 0.5077 | 0.2886 |
| AnnCUR | 0.4724 | 0.4280 | 0.2287 | 0.1919 | 0.2455 | 0.6697 | 0.5842 |
| AnnCUR+KMeans | 0.5083 | 0.4850 | 0.3226 | 0.2517 | 0.3042 | 0.7070 | 0.6184 |
| RBE+KMeans | 0.5431 | 0.4979 | 0.3399 | 0.2539 | 0.3019 | 0.7137 | 0.6300 |
| AnnCUR+Greedy | 0.5618 | 0.5119 | 0.3677 | 0.2960 | **0.3357** | 0.7197 | 0.6565 |
| RBE+Greedy | **0.5849** | **0.5249** | **0.3867** | **0.2992** | 0.3349 | **0.7234** | **0.6682** |

$$binRelevance(q) := \tilde{R}(q, I) \geq q_{1-\frac{K}{|T|}}(R(q, I)),$$

where $K$ is the desired top size and $q_x(v)$ calculates the $x$-th quantile of the vector $v$. We have experimented with various loss functions, but the one described above leads to consistently good results. Further in the experimental section, we will show that this relatively simple approach already gives a visible increase in quality in practice.

## 4.2 SUPPORT ITEMS SELECTION

Following the arguments of Section 3.2, we check various ways of choosing support elements as opposed to the existing approaches that use random selection. All clustering algorithms are taken from the scikit-learn (Pedregosa et al., 2011) library, SpectralClusteringNN is a SpectralClustering with "nearest neighbors" affinity. The algorithms are used with their default parameters because even this simple setting already allows us to significantly improve over the basic solution with random support item selection.

The results of the comparison are presented in Table 2, where the best 3 results for each dataset are highlighted in bold. Clearly, there is a significant superiority of almost any approach based on clustering or diversity over the random selection. The greedy algorithm is the clear winner, second and third places are taken by KMeans and AgglomerativeClustering. However, due to the significantly worse quality of AgglomerativeClustering on the Military dataset, KMeans will be used in further experiments. Another observation is that on the RecSys data, there is a clear superiority of the choice of popular items as the support ones. It is worth mentioning that for RecSys, the elements extracted by popularity are also quite stratified by their categories and an explicit restriction on the number of elements from one category changes the top slightly. However, this may not be true for other data.

## 4.3 NEURAL RELEVANCE-BASED EMBEDDINGS

Following the description in Section 4.1.4, we also apply non-trivial trainable relevance mappings $f_I(R(I, S_Q), \theta_I)$, $f_Q(R(S_I, q), \theta_Q)$ to check whether this modification improves prediction quality

Table 4: Dual encoder embeddings vs support relevances, HitRate, RecSysLT

| Base emb. | Top 100 | Top 200 | Top 300 | Top 400 | Top 500 | Top 600 | Top 700 |
|---|---|---|---|---|---|---|---|
| Dual Encoder Emb. | **0.7048** | **0.6803** | 0.6739 | 0.6739 | 0.6760 | 0.6792 | 0.6827 |
| $R(S_I, q), R(i, S_Q)$ | 0.6300 | 0.6611 | **0.6912** | **0.7109** | **0.7253** | **0.7357** | **0.7448** |

Table 5: Dual encoder embeddings vs support relevances, HitRate($K$, 100), RecSysLT

| Base emb. | K = 100 | K = 200 | K = 300 | K = 400 | K = 500 | K = 600 | K = 700 |
|---|---|---|---|---|---|---|---|
| Dual Encoder Emb. | **0.7048** | 0.7977 | 0.8518 | 0.8855 | 0.9086 | 0.9258 | 0.9385 |
| $R(S_I, q), R(i, S_Q)$ | 0.6300 | **0.8090** | **0.8823** | **0.9190** | **0.9402** | **0.9536** | **0.9629** |

in practice. The results are shown in Table 3. To better interpret the values, the quality of the constant output consisting of popular (in the same sense as in the previous section) elements is also given. It can be seen that in most cases, except for one dataset (Military), trainable relevance mappings improve the final quality of the search for relevant elements and the improvements are obtained for both KMeans and Greedy support elements selection. Note that the transformation that we use is not claimed to be optimal and is given rather to demonstrate that with the help of an easy transformation, one can get an increased quality on various datasets.

## 4.4 DUAL ENCODER EMBEDDINGS VS SUPPORT RELEVANCES

To check whether the embeddings obtained from the relevance vectors are really more useful than other embeddings trained on the service data, we replaced the relevance vectors in the experiments above with embeddings obtained by a dual encoder (the one that is proved to be the best in this task), and trained our algorithm (with KMeans selection of support elements) in the same way, keeping the transformation of the relevance vectors and the training parameters unchanged. The only difference is that, since we used $|S_I| = 100$ requests to a heavy ranker to form an RBE, we must take them into account when calculating the top. Thus, when constructing the top-$X$ output, for dual encoder we use the metric HitRate($X + |S_I|, X$), and for RBE — HitRate($X, X$), which gives the former an advantage with small top sizes.[4] However, starting from $X = 300$, our algorithm is superior to the dual encoder variant, as can be seen from Table 4. A second comparison, where the size of the desired top is fixed, while the number of extracted elements changes, is presented in Table 5. Similarly to the previous comparison, calls to the heavy ranker are taken into account: for the dual encoder, HitRate($K + |S_I|, 100$) is calculated, and for RBE — HitRate($K, 100$). In this comparison, RBE outperforms the dual encoder starting at $K = 200$. Let us note that the actual size of the top used to select candidates before ranking in the production service exceeds the values indicated in the table. A similar comparison with the dual encoder on the data from ZESHEL can be found in Yadav et al. (2022): it is shown that AnnCUR outperforms the dual encoder.

## 5 CONCLUSION

In this paper, we present the concept of Relevance-Based Embeddings. We justify our approach theoretically and show its practical effectiveness on textual (ZESHEL) and recommendation system data. We demonstrate that RBE allows one to obtain better quality in comparison with existing approaches. An important contribution of our work is the study of different strategies for choosing the support elements for RBE. We show that a proper choice of the support elements allows one to significantly boost performance.

Promising directions for future research include a deeper investigation of support element selection strategies as well as applying the proposed RBE to other algorithms, e.g., based on using a heavy ranker during the nearest neighbor search (Morozov & Babenko, 2019).

---

[4]For all sizes of the top, both algorithms were trained once. By training algorithms with different loss functions for each top, taking into account different sizes of tops, the quality of both algorithms can be improved, which, however, is not essential for the current comparison, since the changes will affect both algorithms equally.

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

## A  PROOFS

### A.1  PROOF OF THEOREM 1

**Queries as functions on items and vice versa.**  Each query $q$ defines a function $r_q$ on items: $r_q(i) = R(i, q)$. Let us call two queries $q$ and $q'$ *R-equivalent* if $r_q = r_{q'}$ and write $q \sim_R q'$ to denote this relation. $R$-equivalent queries are interchangeable when it comes to measuring their relevance to any item. Let $Q_R$ be the set of $R$-equivalence classes. $Q_R$ may be considered as an image of $Q$ in $C(I)$ under the mapping $R_Q$ which maps query $q$ to $r_q$. This point of view suggests a natural metric $d_{Q_R}$ on $Q_R$ induced by the uniform norm on $C(I)$: $d_{Q_R}(q, q') = \|r_q - r_{q'}\| = \sup_{i \in I} |R(i, q) - R(i, q')|$.

Now let us note that the map $R_Q : Q \to C(I)$ is continuous since $R$ is continuous and $I$ is compact. Therefore, $Q_R$ is compact as an image of a compact space $Q$ under a continuous mapping. And there is an (injective) embedding of $C(Q_R)$ to $C(Q)$ under which the function $f \in C(Q_R)$ goes to $f \circ R_Q \in C(Q)$. Simply speaking, a continuous function on the equivalence classes of queries is also a continuous function on the queries themselves.

Analogously, we define:

$$R_I : I \to C(Q), \quad R_I(i) = r_i, \quad r_i(q) = R(i, q), \quad R_I(I) = I_R,$$

$$d_{I_R}(i, i') = \|r_i - r_{i'}\| = \sup_{q \in Q} |R(i, q) - R(i', q)|.$$

For convenience, we will identify functions in $C(I_R)$ and $C(Q_R)$ with functions in $C(I_R \times Q_R)$ which are independent of one of their arguments. The relationships mentioned above and similar ones are shown in the diagram below (hooked arrows represent injective mappings, arrows with two heads stand for surjective ones):

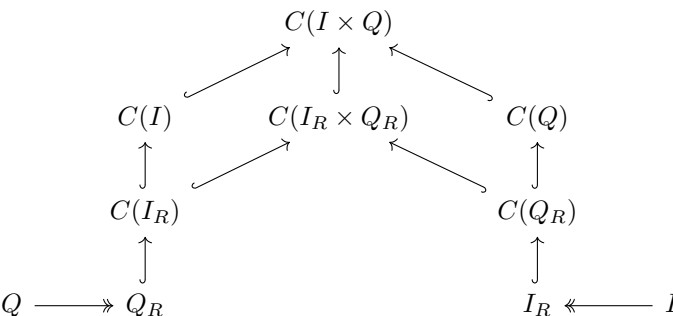

Now, let us make several observations.

**Claim 1.** $r_i \in C(Q_R)$ *and, similarly,* $r_q \in C(I_R)$.

*Proof.* We note that $\|r_i(q) - r_i(q')\| = \|r_q(i) - r_{q'}(i)\| \leq \|r_q - r_{q'}\| = d_{Q_R}(q, q')$. So, $r_i(q) - r_i(q') = 0$ if $r_q = r_{q'}$ and the value $r_i(q)$ does not change if a query is replaced with an equivalent one. It means that $r_i$ is a correctly defined function on the classes of equivalent queries, i.e., on $Q_R$. And finally the same inequality $\|r_i(q) - r_i(q')\| \leq d_{Q_R}(q, q')$ implies that the function $r_i$ is 1-Lipschitz with respect to the metric $d_{Q_R}$. □

**Claim 2.** $R \in C(I_R, Q_R)$.

*Proof.* We have $|R(i, q) - R(i, q')| = |r_q(i) - r_{q'}(i)| \leq \|r_q - r_{q'}\| = d_{Q_R}(q, q')$ and $|R(i, q) - R(i', q)| \leq d_{I_R}(i, i')$. It follows that the value $R(i, q)$ does not change after replacement of a query-item pair $(i, q)$ with some equivalent pair $(i', q')$. So, $R$ can be considered as a function on $I_R \times Q_R$. And by the same inequality $R$ is 1-Lipschitz with respect to the metric $d_R((i, q), (i', q')) = d_{I_R}(i, i') + d_{Q_R}(q, q')$ and hence is continuous. □

**Stone-Weierstrass theorem.** Let us call all functions $r_q$ and $r_i$ *elementary*. Consider the family of all elementary functions $\mathcal{F} = \{r_q | q \in Q\} \cup \{r_i | i \in I\} \subset C(I_R \times Q_R)$.

**Claim 3.** *The family $\mathcal{F}$ separates points in $I_R \times Q_R$, i.e., for each two different points $x, y \in I_R \times Q_R$, there is a function $f \in \mathcal{F}$ such that $f(x) \neq f(y)$.*

*Proof.* Indeed, let $(i_1, q_1)$ and $(i_2, q_2)$ be any two different points in $I_R \times Q_R$. Then $i_1 \neq i_2$ or $q_1 \neq q_2$. Without loss of generality, we can assume that $i_1 \neq i_2$ (they are unequal as points in $I_R$). So, $r_{i_1}$ and $r_{i_2}$ are different functions on $Q_R$ and there exists $q \in Q_R$ such that $r_{i_1}(q) \neq r_{i_2}(q) \Leftrightarrow R(i_1, q) \neq R(i_2, q) \Leftrightarrow r_q(i_1) \neq r_q(i_2) \Leftrightarrow r_q((i_1, q_1)) \neq r_q((i_2, q_2))$. Thus, we found a function ($r_q$) from our family that separates the two points. $\qquad\square$

Next, consider the algebra of functions $\mathbb{R}[\mathcal{F}]$ generated by the family $\mathcal{F}$. This algebra consists of all polynomial combinations of functions in $\mathcal{F}$. More formally, each element of $\mathbb{R}[\mathcal{F}]$ has a representation of the form:

$$\mathbb{R}[\mathcal{F}] \ni f = \sum_{k=1}^{d} c_k \cdot r_{i_{k,1}} \cdot \ldots \cdot r_{i_{k,a_k}} \cdot r_{q_{k,1}} \cdot \ldots \cdot r_{q_{k,b_k}}.$$

In other words, there are $d$ sets $S^1, \ldots, S^d$ of queries and items such that:

$$S^k = S_I^k \cup S_Q^k, \quad S_I^k = \{i_{k,1}, \ldots, i_{k,a_k}\} \subset I, \quad S_Q^k = \{q_{k,1}, \ldots, q_{k,b_k}\} \subset Q.$$

$$f = \sum_{k=1}^{d} c_k \cdot \left( \prod_{i \in S_I^k} r_i \right) \cdot \left( \prod_{q \in S_Q^k} r_q \right). \tag{2}$$

Products of the form $\prod_{i \in S_I^k} r_i$ may be empty and in this case the product equals 1. So, $\mathbb{R}[\mathcal{F}]$ contains constant functions and separates points of $I_R \times Q_R$ (because it contains $\mathcal{F}$). Hence, by Stone-Weierstrass theorem, the algebra $\mathbb{R}[\mathcal{F}]$ is dense in $C(I_R \times Q_R)$. In particular, the function $R$ can be approximated by element of $\mathbb{R}[\mathcal{F}]$ up to an arbitrarily small absolute error.

**Represent polynomials in $\mathbb{R}[\mathcal{F}]$ as products of query and item embeddings.** Consider an arbitrary function $f \in \mathbb{R}[\mathcal{F}]$ and its representation of the form (2). Denote the products $\prod_{i \in S_I^k} r_i(q)$ and $\prod_{q \in S_Q^k} r_q(i)$ by $\pi_{S_I^k}(q)$ and $\pi_{S_Q^k}(i)$ respectively. Consider two $d$-dimensional vectors:

$$e(q) = \left( c_1 \cdot \pi_{S_I^1}(q), \ldots, c_d \cdot \pi_{S_I^d}(q) \right),$$

$$e(i) = \left( \pi_{S_Q^1}(i), \ldots, \pi_{S_Q^d}(i) \right).$$

Then, $f(i, q) = \langle e(i), e(q) \rangle$. Let $S_I = \cup_{k=1}^{d} S_I^k$ and $S_Q = \cup_{k=1}^{d} S_Q^k$. Then $e(q)$ is a continuous (more specifically, polynomial) function of the vector $R(S_I, q)$ and $e(i)$ is a continuous function of the vector $R(i, S_Q)$. So, by the universality theorem for MLPs (Cybenko, 1989; Leshno et al., 1993), the vector $e(i)$ can be approximated up to arbitrarily small absolute error in the form $f_I(R(i, S_Q), \theta_I)$ where $f_I(\cdot, \theta_I)$ — a reach enough MLP architecture. Similarly, $e(q)$ can be approximated by $f_Q(R(S_I, q), \theta_Q)$. Hence, $\langle f_I(R(i, S_Q), \theta_I), f_Q(R(S_I, q), \theta_Q) \rangle$ approximates $f(i, q)$. Finally, we can consider $f \in \mathbb{R}[\mathcal{F}]$ such that $\|f - R\| < \frac{\varepsilon}{2}$ and then find such $\theta_I$ and $\theta_Q$ that $\|f - \langle f_I(R(i, S_Q), \theta_I), f_Q(R(S_I, q), \theta_Q) \rangle\| < \frac{\varepsilon}{2}$. These parameters will give us a desired $\varepsilon$-approximation of $R$ in a form of product of relevance-based embeddings.

### A.2 Proof of Corollary 1

Let us take some $\frac{\varepsilon}{2}$-approximation of $R$ of the form

$$R(i, q) \approx \langle e_I(q), e_Q(i) \rangle = \langle f_Q(R(S_I, q), \theta_Q), f_I(R(i, S_Q), \theta_I) \rangle$$

via the relevance-based embeddings $e_I(q)$ and $e_Q(i)$ of dimension $d$. Take a constant $C$ such that $\|e_I(i)\| < C$ and $\|e_Q(Q)\| < C$ for all $q \in Q$ and $i \in I$ and consider the following vector embeddings of dimension $d + 2$:

$$\tilde{e}_I(i) = \left( \frac{1}{C} e_I(i), \sqrt{1 - \|\frac{1}{C} e_I(i)\|^2}, 0 \right),$$

$$\tilde{e}_Q(q) = \left( \frac{1}{C} e_q(q), 0, \sqrt{1 - \|\frac{1}{C} e_q(q)\|^2} \right).$$

Note that $\langle e_I(i), e_Q(q) \rangle = C^2 \cdot \langle \tilde{e}_I(i), \tilde{e}_q(q) \rangle$ and $\tilde{e}_I(i)$ and $\tilde{e}_q(q)$ lie on the $(d+1)$-dimensional unit sphere $S^{d+1} \subset \mathbb{R}^{d+2}$. We can take new powerful enough architectures $\tilde{f}_I$ and $\tilde{f}_Q$ with outputs normalized to unit sphere in $\mathbb{R}^{d+2}$ and fit for them parameters $\tilde{\theta}_I$ and $\tilde{\theta}_Q$ such that $\tilde{f}_I(R(i, S_Q), \tilde{\theta}_I) \approx \tilde{e}_I(i)$ and $\tilde{f}_Q(R(S_I, q), \tilde{\theta}_Q) \approx \tilde{e}_Q(q)$ and $\tilde{R}(i, q) = \langle \tilde{f}_I(R(i, S_Q), \tilde{\theta}_I), \tilde{f}_Q(R(S_I, q), \tilde{\theta}_Q) \rangle \approx \langle \tilde{e}_I(i), \tilde{e}_Q(q) \rangle$. More specifically take $\tilde{\theta}_I$ and $\tilde{\theta}_Q$ such that:

$$|\langle \tilde{e}_I(i), \tilde{e}_Q(q) \rangle - \tilde{R}(i, q)| < \frac{\varepsilon}{2C^2} \Rightarrow$$

$$|\langle e_I(i), e_Q(q) \rangle - C^2 \tilde{R}(i, q)| < \frac{\varepsilon}{2}$$

Given that $|R(i, q) - \langle e_I(i), e_Q(q) \rangle| < \frac{\varepsilon}{2}$ it yields $|R - a\tilde{R}| < \varepsilon$. Which means that statement of corollary is satisfied with $a = C^2$.

## B  GREEDY SELECTION OF SUPPORT ITEMS

It can be shown that the CUR approximation replaces every item with a linear combination of support items so that the MSE between the true relevances and their CUR approximations on the train set of queries is minimized. Thus, our goal is to choose support items so that the MSE error after the CUR decomposition is minimal. Formally we need to solve the following problem:

*Optimal CUR-decomposition*: Assume that we are given an $n \times m$ matrix $X$ of real numbers and let $x_i$, $i = 1, \ldots, n$, be the rows of $X$. Choose $k$ rows in such a way that the sum of squared distances from each row of the matrix to the space generated by the chosen rows would be minimal. In other words find subset of indices $S = \{i_1, \ldots, i_k\} \subset \{1, \ldots, n\}$ which minimizes following expression:

$$\sum_{i=1}^{n} \|x_i - \pi(x_i, span(x_{i_1}, \ldots, x_{i_k}))\|_2^2 = \sum_{i=1}^{n} \|x_i - X_S^T \text{pinv}(X_S^T) x_i\|_2^2,$$

where $X_S$ is an $k \times m$ matrix consisting of rows with indices from $S$. This problem corresponds to the CUR-decomposition of $X$ with $k$ rows and all $m$ columns.

A straightforward way is to choose items greedily. Suppose we have already chosen items $i_1, \ldots, i_t$. Then, we choose an item $i_{t+1}$ so that

$$\sum_{i=1}^{n} \|x_i - \pi(x_i, span(x_{i_1}, \ldots, x_{i_{t+1}}))\|_2^2$$

is minimal.

Let us discuss how to choose $x_{i_{t+1}}$. Let $\Delta^t$ be the $n \times m$ matrix of our current approximation errors: $\Delta_i^t = x_i - \pi(x_i, span(x_{i_1}, \ldots, x_{i_t}))$ ($\Delta^0 = X$). Note that $span(x_{i_1}, \ldots, x_{i_t}, x_i) = span(x_{i_1}, \ldots, x_{i_t}, \Delta_i^t/\|\Delta_i^t\|_2)$, so for the purpose of evaluation our objective we can replace $x_i$ with $o_i^t = \Delta_i^t/\|\Delta_i^t\|_2$. When we add $x_i$ to the support set, the squared error on $x_j$ reduces by $\langle x_j, o_i \rangle^2$ and $\Delta_j$ becomes $\Delta_j - \langle x_j, o_i \rangle o_i$. It can be seen by considering the orthonormal basis of $\mathbb{R}^m$, the first $t$ elements of which generate $span(x_{i_1}, \ldots, x_{i_t})$ and $(t+1)$-th is $o_i^t$. Adding $o_i^t$ to the

support set will set to zero the $(t+1)$-th coordinate of the vector $x_j^t$ (and $\Delta_j^t$). And in the standard basis this coordinate may be calculated as $\langle x_j, o_i \rangle$. So we want to maximize over $i$:

$$\sum_{j=1}^n \langle x_j o_i^t \rangle^2 = \sum_{j=1}^n o_i^{tT} x_j x_j^T o_i^t = o_i^{tT} \left( \sum_{j=1}^n x_j x_j^T \right) o_i^t = o_i^{tT} X^T X o_i^t.$$

Thus, the algorithm is as follows:

---

**Algorithm 1** $l_2$-greedy support items selection

---

compute $X^T X$
compute normalized vectors $o_i^0 = x_i / \|x_i\|_2$
$N \leftarrow n$
**for** $t$ in $[1, \dots, k]$ **do**
    choose $i_{t+1}$ which maximize $o_i^{tT} X^T X o_i^t$
    update all $o_j$
    **for** j in $[1, \dots n]$ **do**
        $o_j^{t+1} \leftarrow o_j^t - o_{i_{t+1}}^t \langle o_{i_{t+1}}^t, o_j^t \rangle$
        $o_j^{t+1} \leftarrow o_j^{t+1} / \|o_j^{t+1}\|_2$
    **end for**
**end for**

---

The choice of the next support item may be trivially implemented with $O(m^2 n)$ complexity. But it can be optimized: together with $o_j^t$ we can keep the vectors $c_j^t = X^T X o_j^t$ that can be computed once initially in $O(m^2 n)$ and can be updated at each iteration synchronously with $o_j^t$. Updates of $o_j^t$ at each iteration have the form $o_j^{t+1} = \alpha o_j^t + \beta o_{i_{t+1}}^t$, so $c_j$ transforms analogously with the same coefficients: $c_j^{t+1} = \alpha c_j^t + \beta c_{i_{t+1}}^t$. So we can score all the items in $O(mn)$, calculating all the dot products $\langle o_j^t, c_j^t \rangle$ and update vectors $o_j$ and $c_j$. The total complexity of the algorithm is $O(mn(m+k))$.

