# OpenReview forum: "Relevance-based embeddings for efficient relevance retrieval"
_ICLR.cc/2024/Conference — Submitted to ICLR 2024_

### Official Review · Reviewer_ayPq · 2023-10-30

**Soundness:** 3 good
**Presentation:** 3 good
**Contribution:** 2 fair
**Rating:** 6
**Confidence:** 3

**Summary:**

This paper proposes a novel approach to ML-based query matching: instead of the classical "fast-and-cheap-retrieval followed by high-quality re-ranking," the authors propose leveraging the more expensive re-ranking model to create relevance-based embeddings. This approach describes each query by its relevance wrt a set of support elements, and the authors investigate several strategies for selecting these support items and show empirically that such a strategy leads to significant improvements.

**Strengths:**

The problem tackled in this paper (i.e., efficient relevant retrieval) has strategic practical applications, from Information Retrieval to Recommender Systems to Question Answering.

The approach introduced in this paper appears to be original: while building on pre-existing work [Yadav et al 2022] [Morozov & Babenko, 2019], the authors introduce an original, principled, non-random way to identify the support elements for the computation of the Relevance-Based Embeddings (RBE). Their experiments show that the novel approach outperforms previous work in this field.

The paper is reasonably well-written and organized, which makes is fairly easy to follow.

**Weaknesses:**

The paper can be most improved by clarifying and beefing up the Empirical Section:
- the 7 datasets that you are using are extremely small in size (the largest has only 104K items)
- related to the issue above, please explain whether (and how) your approach can scale to 10 B items (i.e., up to 5 orders of magnitude)
- you should add as additional datasets at least one large-size Question Answering domain; this would show that (i) your approach can scale, and (ii) your approach also applies to one of the most-studied problems of the last few decades

OTHER COMMENTS:
- to increase the readability of Table 2, please color-code the top-1/2/3 results (eg, Red, Green, Blue)
- in the caption of Table 4, please specify what metric do the numbers represent (like you din in Table 2, 3, and 5)
- your last paragraph of the intro is very weak. Please beef-it up and quantify your statements with an intuitive summary of your main empirical results
- you should take 3-4 sentences to intuitively explain Figure 1. In the current draft, you (wrongfully) refer to it as "Fig 3.1" , and you do not offer any narrative explaining (the intuition behind)MLP it.
- in the intro, to make the paper easier to follow by non-specialist readers, it would be nice to add 3-4 sentences to discuss the intuition behind dual-/cross- encoders and the types of features that each of them can use (and why)
- please add a reference to the page 1 statement "the cross-encoder ones are generally more powerful"
- before using an acronym for the first time, please use the full-name (eg, on top of page 2, with the use of "MLP")
- a few language issues:
   - p 1: "straightforward search is unacceptable" --> "brute-force/exhaustive search is not feasible"
   - p 2: "exchange of complexity for quality" --> "trading-off complexity for quality"
   - p 5: "films" --> "movies"
   - p 8: "with the exception of one dataset" --> "with the exception of one dataset (Military)"
   - multiple places: "real" --> "real-world"

**Questions:**

Please explain why are you using only 5 of the 16 domains in ZESHEL

---

> ### Author Response · Authors · 2023-11-15
> **Authors' response**
>
> Thank you for your review and suggestions!
>
> **W: The 7 datasets that you are using are extremely small in size (the largest has only 104K items)**
>
> And
>
> **W: You should add as additional datasets at least one large-size Question Answering domain; this would show that (i) your approach can scale, and (ii) your approach also applies to one of the most-studied problems of the last few decades**
>
> We agree that adding a larger dataset would be useful for the paper. We are working on adding the MsMarco dataset, as suggested by Reviewer UmnS. However, there is a question of whether we will manage it during the discussion period since time limitations are quite challenging.
>
> **W: Related to the issue above, please explain whether (and how) your approach can scale to 10 B items (i.e., up to 5 orders of magnitude)**
>
> Let us consider separately the selection of the key elements, training and inference:
>
> ### Key Selection:
> Since different clusterization approaches have shown near-optimal quality, there are different options for scalable clustering:
>
> - Clusterization on downsampled datasets: for extremely large (and dense) datasets it is natural to expect that cluster structure could be inferred from a significantly smaller subsample (according to the authors' experience, on the data of ads recommendation systems with billions of banners and downsampling to millions, this is so)
>
> Even for our research (small) datasets downsampling 75% of data reduces the quality of key selection in a discussable way:
>
> | key | Yugioh | P.Wrest. | StarTrek | Dr.Who | Military |
> |---|---|---|---|---|---|
> | AnnCUR+Random | 0.4724 | 0.4280 | 0.2287 | 0.1919 | 0.2455 |
> | AnnCUR+KMeans | 0.5083 | 0.4850 | 0.3226 | 0.2517 | 0.3042 |
> | AnnCUR+1/4KMeans | 0.5112 | 0.4685 | 0.3101 | 0.2514 | 0.2854|
>
> We are going to provide a detailed trade-off in the supplemental materials.
>
> - Using data-driven clusterization: as shown in the third row of Table 2, choosing popular items from different categories/genres (in our case the global top of popular items is almost uniformly diversified) works extremely well.
>
> - Using a distributed clustering algorithm
>
> ### Training:
> As discussed in Sections 3.3 and 4.1.4, the training of such a model does not significantly differ from any dual-encoder-like models (which are commonly used in production recommendation services) with relevance vectors as inputs. The only major difference is that relevances to fixed support items should be provided. Let us also note that efficient sampling of negative samples for the loss function should be used in order to train on such large datasets.
>
> ### Inference:
> It is also similar to dual-encoders: the item representations are precalculated and placed in the Approximate Nearest Neighbours index like HNSW, which accepts the embedding of the query as input.
> The other comments are about readability, text and language issues. We will correct the corresponding parts in the revised text. Thank you for the extremely detailed feedback, it really allows you to make the text better!
>
> **Q:Please explain why are you using only 5 of the 16 domains in ZESHEL**
>
> In our experiments on public datasets, we follow the setup of Yadav et al. (2022) and use the same datasets - please, see Section B.3 of their paper.
>
> We hope that our response addresses the concerns.

---

> ### Author Response · Authors · 2023-11-15
> **Question**
>
> We just noticed that the initial rating changed from 8 while we were writing the answers, although the review remained the same - we would be very grateful if you could clarify the motivation.

---

> > ### Comment · Reviewer_ayPq · 2023-11-16
> > **Change of rating**
> >
> > After reviewing the comments and concerns of the other reviewers, I have changed my original rating to reflect issues that I have missed. I am looking forward to your replies, which will be carefully considered towards the final rating.

---

> > > ### Author Response · Authors · 2023-11-17
> > >
> > > Thank you for your reply! Please note that we replied to your concerns in our [comment above](https://openreview.net/forum?id=mssRRt6OPE&noteId=7zPc93IF7X). If you have any additional questions - we look forward to further discussions.

---

> ### Author Response · Authors · 2023-11-22
> **Authors' response**
>
> Thank you again for your detailed feedback! We updated the paper accordingly. This [comment](https://openreview.net/forum?id=mssRRt6OPE&noteId=juOzSsyvNV) summarizes our updates. We also conducted the experiments on the MsMarco dataset, the results are consistent with those reported in the paper, see this [comment](https://openreview.net/forum?id=mssRRt6OPE&noteId=zTI5Srl133) for the details.
>
> We will be happy to hear your thoughts regarding our reply and updates.

---

### Official Review · Reviewer_9wpg · 2023-10-31

**Soundness:** 3 good
**Presentation:** 2 fair
**Contribution:** 3 good
**Rating:** 6
**Confidence:** 4

**Summary:**

This paper is in the retrieval-and-rank setting where, in practice, there are usually a low-cost retrieve stage to get a subset of good items, followed by a reranking stage to use complicated method for fine-grained ranking. The paper proposes to build embedding for query and item based on the later reranking stage, and leverage that for the first retrieval stage. There are theories proved to justify the soundness, followed by some experiment results showing the effectiveness of the proposed method.

**Strengths:**

I believe 5-10 years ago, even embedding based KNN is considered slow and is not used in the retrieval stage. Instead people use locality Hashing or approximate KNN. Now there are fundamental advancement in terms of embedding based KNN acceleration, and this paper's topic becomes very important. Given we need to do a sophisticated reranking, is it possible to use embedding to approximate. This will largely help with the performance of the recommendation or information retrieval task. In addition to that, the theory is sound in the paper. One extension could be about the estimation of the dimension needed for a tolerance \epsilon, but that will bring in a lot of difficulties. The writing before section 4 is also informative and clear. Overall I believe it's a good paper.

**Weaknesses:**

How to make the work practical seems to be the number one weakness. Though it's proved that such decomposition exists, finding those embedding mapping for Q and I can be very difficult. I'm looking for a more rigorous way to do that. For example, how to set the right dimension and have the confidence that the dimension is good enough the meet the approximation error \epsilon. How to set the right \epsilon and knowing it's small enough to ensure the retrieval quality is better than existing method e.g. from a set of embedding learned just for the retrieval stage.

In addition, I don't think "relevance" matters much in the context of the paper. Rewriting the paper into the standard retrieve-and-rank language would help. The problem is essentially how to improve the coarse retrieve stage by leveraging the rank stage model. The latter doesn't have to be a relevance model, but rather any fine-grained reranking model. The paper mentioned some examples like QA, which also doesn't fall into the relevance setting. One of the most popular use cases is to concatenate Q and I for transformer-based model to handle, and it's also not necessarily a relevance model.

In the experiment part, the latency or other system related metrics should be reported as well. The number of items retrieved could be tuned based on the retrieval stage quality as well as the system constraint. Directly comparing the proposed embedding with other baselines may not be helpful. They can easily be used to retrieve different number of items before reranking. Given the two-stage setting is mainly for computational cost concern (otherwise one can just use the reranking model for all Q-I pairs), the computation/system metrics are worth checking and reporting.

The writing in section 4 seems much worse compared to the previous sections. There are details or intuitions lacking (potentially due to page limit). Also adding multi-modal datasets e.g. video-language retrieval could help strengthen the paper.

**Questions:**

Please see the weakness section

---

> ### Author Response · Authors · 2023-11-15
> **Authors' response**
>
> Thank you for your review and suggestions!
>
> > How to make the work practical seems to be the number one weakness. Though it's proved that such decomposition exists, finding those embedding mapping for Q and I can be very difficult. I'm looking for a more rigorous way to do that. For example, how to set the right dimension and have the confidence that the dimension is good enough the meet the approximation error \epsilon. How to set the right \epsilon and knowing it's small enough to ensure the retrieval quality is better than existing method e.g. from a set of embedding learned just for the retrieval stage.
>
> In practice, the overall time limitations (i.e., the CE call budget, since the overall time complexity depends on it) are set externally and are determined, for example, by the response timeout of a recommendations service.
> Given the overall CE call budget, the number of key items could be selected similarly to any other hyperparameters like the dual encoder embedding size. In our experiments, we fixed the embedding size to 100 which already gives performance improvements.
>
> > Rewriting the paper into the standard retrieve-and-rank language would help.
>
> Thank you for the suggestion, we can add such an interpretation to the text or use this terminology if required. Could you, please, specify, what terminology you suggest (i.e., what do you think we should use instead of "queries", "items", and "relevance").
>
> > In the experiment part, the latency or other system related metrics should be reported as well. The number of items retrieved could be tuned based on the retrieval stage quality as well as the system constraint. Directly comparing the proposed embedding with other baselines may not be helpful. They can easily be used to retrieve different number of items before reranking.
>
> In both ZESHEL and RecSys data, t(heavy ranker/cross encoder call) > t(dual encoder part call) >> t(dot product or tiny $f_Q$, $f_I$ MLPs).
> The item/document representations for AnnCUR/RBE and the dual encoder are offline-precalculated and do not cost any additional time per query.
> Once the dual encoder or AnnCUR/RBE query part is calculated, the calculation of relevance for any item is just a dot product calculation, which is much faster than one CE call.
>
> Finally, if we have K items on the final stage of ranking, the overall time consumption is equal to:
>
> - K CE calls
> - 1 DE/RBE/AnnCUR query representation calculation: for RBE and AnnCUR it includes several CE calls that we take into account in our comparisons
> - Dot product calculations that are altogether significantly cheaper than CE calls for any reasonable K (~hundreds-thousands)
>
> That's why we and the authors of AnnCUR measure the overall performance in terms of “fixed CE calls budget”. For our Tables 2-3, we compare the methods with the same CE calls (top size). For our Tables 4-5, we also evaluate both methods with the same CE calls: the top size for DE is larger since RBE uses 100 calls per query for the relevances calculation.
> We could provide the exact timings for each model, but in both cases the overall complexity of recommendations will be equal to the CE calls time + eps.
>
> > The writing in section 4 seems much worse compared to the previous sections. There are details or intuitions lacking (potentially due to page limit).
>
> We will update this section in the revised version of the paper.
>
> > Also adding multi-modal datasets e.g. video-language retrieval could help strengthen the paper.
>
> Let us note that the items in RecSys/RecSysLT datasets are described by texts, table data and images. All these sources of information are used as inputs for both DE and CE. If we misunderstood your comment, please correct us.

---

> ### Author Response · Authors · 2023-11-22
> **Authors' response**
>
> Please note that we updated the paper according to the comments from the reviewers. This [comment](https://openreview.net/forum?id=mssRRt6OPE&noteId=juOzSsyvNV) summarizes our updates. We also conducted the experiments on the MsMarco dataset, the results are consistent with those reported in the paper, see this [comment](https://openreview.net/forum?id=mssRRt6OPE&noteId=zTI5Srl133) for the details.
>
> We will be happy to hear your feedback regarding our reply and updates.

---

### Official Review · Reviewer_NsRC · 2023-10-31

**Soundness:** 4 excellent
**Presentation:** 3 good
**Contribution:** 3 good
**Rating:** 5
**Confidence:** 3

**Summary:**

This paper investigates relevance-based embeddings. The authors employ support items to construct relevance vectors before training an embedding on top of them. This approach preserves the efficiency advantages of the dual-encoder model in comparison to the cross-encoder and enhances the performance based on previous test results for dual-encoder models. Unlike previous work, such as AnnCUR, which also utilizes the concept of support items, this paper generalizes the embedding format, achieves superior test accuracy, and additionally explores the selection of support items.

**Strengths:**

1. An innovative method to enhance the performance of the dual-encoder while preserving its efficiency.
2. A generalization of AnnCUR’s approach to using support items, accompanied by proofs of the method’s expressiveness.
3. A comprehensive study examining the impact of various choices of support items.
4. A clear description of the method and experiments conducted.

**Weaknesses:**

1. I am concerned that the experiments conducted on the author's method may only be effective in scenarios such as "entity linking," or similar straightforward datasets characterized by a clear clustering structure based on entities. Is it feasible to apply the author's method to different types of datasets, such as "question-answer datasets"?
2. While the author's method does extend the capabilities of AnnCUR, the performance improvement shown in the table from AnnCUR+KMeans to RBE+KMeans is somewhat modest, with an increase of less than 2 percent in most columns except for the first one.
3. Although efficiency is a noted advantage of dual-encoder based methods, employing a large set of support items can also lead to substantial computational complexity. Would it be possible to conduct a comparison of efficiency between the author's method, dual-encoder, and cross-encoder, along with a comparison of their respective accuracies? I believe that such a comparison of efficiency is crucial.

**Questions:**

1. At the conclusion of page 6, I found myself confused about the necessity for concatenating R(SI, q) with F(R(SI, q), θ). What is the rationale behind not solely utilizing the second term? How does this tie into your previous statement regarding the examination of whether such vector transformations enhance the quality of predictions?
2. Regarding Theorem 1, is there any limitation on the sizes of SQ and SI? The guarantee of expressivity may become insignificant if SQ or SI encompasses the entire dataset.

---

> ### Author Response · Authors · 2023-11-15
> **Authors' response (1/2)**
>
> Thank you for your review and suggestions!
>
> **W: Is it feasible to apply the author's method to different types of datasets, such as "question-answer datasets"?**
>
> Since we applied the method to a recommendation task (RecSys/RecSysLT datasets), it seems to be suitable for any similar information retrieval task. Moreover, we are working on adding the MsMarco dataset suggested by Reviewer UmnS. However, there is a question of whether we will manage to finalize it during the discussion period. Otherwise, we will add the results to the camera-ready version.
>
> **W: …the performance improvement shown in the table from AnnCUR+KMeans to RBE+KMeans is somewhat modest…**
>
> First, let us note that AnnCUR+KMeans is already a strong baseline - the original AnnCUR uses the random selection of key items and we show that it can be significantly improved, which is an important part of our contribution (together with the theoretical analysis). When comparing the proposed method with previous works (i.e., with AnnCUR), we see that, e.g., on the Star Trek dataset, RBE+KMeans has almost 1.5 times greater performance than AnnCUR (0.23 -> 0.34 HiteRate), on other datasets the improvements are also noticeable.
>
> We add AnnCUR+KMeans to our experiments in order to show that both our improvements (selection of key items and aggregation of their relevance using $f_Q$, $f_I$) give some profit. As we write on page 6, the purpose of the experiments with relevance aggregation is to show that even the use of trivial transformations $f_Q$ and $f_S$ allows one to get some profit. Since we use really simple MLP-based transformations and do not aim to find the best practical option (over a proof-of-concept), we consider this result acceptable. Some options for increasing this profit in practical applications are described in Section 3.3, but since they are more practical than theoretical and depend significantly on the data set and subject area, we have not experimented with them.
>
> **W: Although efficiency is a noted advantage of dual-encoder based methods, employing a large set of support items can also lead to substantial computational complexity.  Would it be possible to conduct a comparison of efficiency between the author's method, dual-encoder, and cross-encoder, along with a comparison of their respective accuracies?**
>
> In both ZESHEL and RecSys data, t(heavy ranker/cross encoder call) > t(dual encoder part call) >> t(dot product or tiny $f_Q$, $f_I$ MLPs).
>
> The item/document representations for AnnCUR/RBE and the dual encoder are offline-precalculated and do not cost any additional time per query.
>
> Once the dual encoder or AnnCUR/RBE query part is calculated, the calculation of relevance for any item is just a dot product calculation, which is much faster than one CE call.
>
> Finally, if we have K items on the final stage of ranking, the overall time consumption is equal to:
>
> - K CE calls
> - 1 DE/RBE/AnnCUR query representation calculation: for RBE and AnnCUR it includes several CE calls that we take into account in our comparisons
> - Dot product calculations that are altogether significantly cheaper than CE calls for any reasonable K (~hundreds-thousands)
>
> That's why we and the authors of AnnCUR measure the overall performance in terms of “fixed CE calls budget”. For our Tables 2-3, we compare the methods with the same CE calls (top size). For our Tables 4-5, we also evaluate both methods with the same CE calls: the top size for DE is larger since RBE uses 100 calls per query for the relevances calculation.
>
> We could provide the exact timings for each model, but in both cases the overall complexity of recommendations will be equal to the CE calls time + eps.

---

> ### Author Response · Authors · 2023-11-15
> **Authors' response (2/2)**
>
> **Q: At the conclusion of page 6, I found myself confused about the necessity for concatenating R(SI, q) with F(R(SI, q), θ). What is the rationale behind not solely utilizing the second term?**
>
> The initial intuition was that the <$f_I$(...), $f_Q$(...)> expression splits into two terms - the prediction of AnnCUR and the trainable prediction of its error (thank you for pointing out that we forgot to specify this in the text - it will be added). In the experiments, such additions improved the convergence and training stability. Moreover, since we have used $|S_I$| CE calls to calculate R(S_I, q), the computational complexity of this term is insignificant.
>
> As you suggested, we trained RBE without the first term on the RecSys dataset and achieved better test quality (~ +0.01 HR, all other parameters are the same), so, in some cases, it may even improve the quality. We will try to add a similar comparison for other datasets in the final version of the paper.
>
> Note that many other improvements can be proposed here, but, as mentioned earlier, our goal was to show the possibility of making a profit for the idea as a whole, and not to go into the search for various modifications of the architectures allowed here.
>
> **Q:  Regarding Theorem 1, is there any limitation on the sizes of SQ and SI? The guarantee of expressivity may become insignificant if SQ or SI encompasses the entire dataset.**
>
> As mentioned in Section 3.3, there are practical examples, where the query set Q is infinite - text search or any recommendation tasks with floating number factors in query/user. Under this setting, the result is non-trivial. We need more time to think about estimates of the sizes, if we come up with some new results - we will extend our reply.

---

> > ### Comment · Reviewer_NsRC · 2023-11-20
> > **Thank you for your reply**
> >
> > Thank you for your reply. It resolved some of my confusion. I appreciate the authors' solid experiments with different choices of support items, and I agree that switching from random support items to KMeans can significantly improve performance, as shown in the table. However, concerning the REB method, I'm sorry to say that I'm not convinced by its advantages as demonstrated in the experiment on page 7.

---

> > > ### Author Response · Authors · 2023-11-22
> > > **Authors' response**
> > >
> > > Thank you for engagement in the discussion. Let us clarify that we consider the selection of support elements to be a part of the proposed relevance-based embeddings: the relevance vectors clearly depend on this choice.
> > >
> > > We also updated the paper according to the comments from all the reviewers, this [comment](https://openreview.net/forum?id=mssRRt6OPE&noteId=juOzSsyvNV) summarizes our changes. In particular, in the updated version, we add one more approach for selecting the support items that further improves the performance. Importantly, we still see the improvements from trainable relevance mappings. We also conducted the experiments on the MsMarco dataset, the results are consistent with those reported in the paper, see this [comment](https://openreview.net/forum?id=mssRRt6OPE&noteId=zTI5Srl133) for the details. Again, the best results are achieved with trainable relevance mappings.

---

> > > > ### Comment · Reviewer_NsRC · 2023-11-22
> > > >
> > > > Thank you for providing an update on the MsMacro experiment results. I agree that the author's method performs well on various datasets.

---

### Official Review · Reviewer_UmnS · 2023-11-09

**Soundness:** 3 good
**Presentation:** 3 good
**Contribution:** 2 fair
**Rating:** 6
**Confidence:** 3

**Summary:**

In this paper, the authors have introduced a novel approach for generating embeddings based on the relevance between query-item sets and item-query sets, which they refer to as Relevance-Based Embedding (RBE). Unlike traditional methods that focus on individual query-item relevance pairs, RBE represents a query with a relevance score based on a defined set of support items, and conversely, it represents items with relevance scores on a set of support queries.

Moreover, the paper also presents strategies for the selection of these support items and queries.

**Strengths:**

Strength:
1. A Novel approach for query and item representation.
2. Experimental analysis on 7 real world text and recommendation datasets.

**Weaknesses:**

Comparison with matrix factorizations: The RBE approach is a collaborative filtering recommendation approach. The relevance score collaborative filtering has traditional methods like matrix factorization (low norm decomposition) and mixed filtering approaches like Factorization machines. How does RBE compare to them? It will be nice to have a detailed differentiation and comparison between RBE and matrix factorization.

**Questions:**

1. How does RBE based embedding works for ranking in MsMarco query-passage/document datasets?
2. There can be certain tail queries that are highly relevant to non support items. How does their representation get affected?

---

> ### Author Response · Authors · 2023-11-15
> **Authors' response**
>
> Thank you for your review and suggestions!
>
> **Q: There can be certain tail queries that are highly relevant to non support items. How does their representation get affected?**
>
> To check the performance on such tail queries, we explicitly sorted the test queries from the RecSysLT dataset by their maximal relevance to key items (selected by the KMeans algorithm) and measured the quality on four different parts of the test data. The obtained results for different algorithms are presented in the table below (HitRate@100 on subsample / HitRate@100 on the entire test sample * 100%):
>
> | | q0...q25 | q25...q50 | q50...q75 | q75...q100 |
> |---|:---:|:---:|:---:|:---:|
> |Popular| -7,6%| 7,19%| 7,03% | -6,62%|
> ||| -0,21% | 0,21% ||
> |AnnCUR+KMeans | -1,25% | -0,48% | 3,06% | -1,34%|
> ||| -0,86% | 0,86% ||
> | Dual Encoder | -2,94% | -3,63% | 6,65% | -0,08%|
> ||| -3,28% | 3,28% ||
> | RBE+KMeans | -7,95% | 0,88% | 5,95% | 1,12%|
> ||| -3,53% | 3,53% ||
>
> Note that for the more relevant half, the quality is better for all algorithms, including Dual Encoder, which does not use key elements in any way. This may be a consequence of the fact that the less relevant half is represented by those queries that we generally know less about and thus give worse recommendations. Another observation is that in the best quarter, the quality of 3 algorithms becomes worse than average, which is an unexpected result.
>
> We hope these results address the question and we plan to add a similar analysis for all the datasets to the supplemental material.
>
> **Q: How does RBE based embedding works for ranking in MsMarco query-passage/document datasets?**
>
> Thank you for this suggestion. We are currently working on additional experiments with this dataset. This would take some time, we hope that it will be finished during the author feedback period, but otherwise we will add the results to the camera-ready version of the paper.
>
> **Q: It will be nice to have a detailed differentiation and comparison between RBE and matrix factorization.**
>
> Most matrix factorization methods (for example, the matrix factorizations of user-to-item ratings matrix) could not operate with new queries/users effectively since new rows require factorization of this new matrix which is not applicable to online recommendations, while RBE can handle new queries since it the relevances to key items (according to the heavy ranker) can be evaluated online.
>
> We could evaluate a matrix factorization method by masking all relevances of test queries except for the key (support) items and trying to reconstruct the rest. Do you think that this experiment will be helpful for supporting our claims?
>
> If we misunderstood your question, please correct us.

---

> ### Author Response · Authors · 2023-11-22
> **Authors' response**
>
> Thank you for suggesting the MsMarco dataset. We conducted preliminary experiments on this dataset and plan to add the complete results to the final version of the paper. The results are consistent with those reported in the paper, see this [comment](https://openreview.net/forum?id=mssRRt6OPE&noteId=zTI5Srl133) for the details. We also updated the paper according to the comments of the reviewers, this [comment](https://openreview.net/forum?id=mssRRt6OPE&noteId=juOzSsyvNV) summarizes our updates. We will be happy to hear your feedback regarding our reply and updates.

---

### Author Response · Authors · 2023-11-22
**Revised paper**

We would like to thank the reviewers for their valuable comments. We updated the paper accordingly. In particular, the following modifications have been made:

- Following our discussion with the reviewers, we added a paragraph about scalability to Section 3.3.

- Some other textual improvements have been made: in particular, we added more details to the introduction, added details and motivation to Section 4.1.4., the typos have been corrected.

- Additionally, we extended the list of simple strategies for selecting support elements with a greedy approach that optimizes the MSE error of the approximation of the relevance matrix, see Section 3.2. As one may expect, this approach outperforms other heuristics and thus improves the performance even further.

As can be seen from the updated Table 3, the improvements obtained by trainable relevance mappings are consistent: similarly to our previous experiments with KMeans, we obtain improvements in combination with Greedy.

If there are any further suggestions for improving the paper, we will be happy to address them.

---

### Author Response · Authors · 2023-11-22
**Additional experiments with MsMarco**

To address the request for additional experiments with the MsMarco dataset, we conducted the following preliminary experiment. We took the dataset prefix of 3K test queries and 24K passages corresponding to them. As a heavy ranker, we used a model from the [SentenceTransformers library](https://huggingface.co/sentence-transformers/all-mpnet-base-v2), which is trained, among other datasets, on the MsMacro data.

Next, we reproduced our experiment from Tables 2 and 3 in a similar way on a subset of the dataset (rows are not mixed, i.e., the queries from the test are not present in the training).

The following results are obtained:
| | HitRate@100 |
|----------|----------|
|AnnCur (Baseline) | 0.6130 |
| RBE + KMeans | 0.6205 |
| AnnCUR + Greedy | 0.6459 |
| RBE + Greedy | 0.6564 |

We see that the results support the conclusions of our experiments.

In the final version of the paper, we will present the experiment on the full dataset. If the reviewers have additional suggestions on how we can improve the pipeline of our experiment, we will be happy to take them into account in the final version.

---

### Author Response · Authors · 2023-11-23
**The end of the discussion period**

The author-reviewer discussion period is closing now and unfortunately, we have not received feedback from three of the reviewers. Thus, if there are any further questions or comments, we may not be able to reply. However, we hope that we addressed all the concerns with our replies, updates of the text, and several additional experiments that support our claims.

Sincerely,
Authors

---

### Meta-Review · Area_Chair_2Tfs · 2023-12-11

**Metareview:**

This paper presents a method to improve the embedding methods for retrieval purpose. The overall sentiment of the paper is okay, but not super exciting.

Strength:
1. An interesting method to enhance the performance of the dual-encoder with supported items using information from the later ranking stage.
2. A comprehensive evaluation on several datasets.

Weakness:
1. All datasets studied in this paper are very small, even with MS Marco, making it difficult to assess the practical usefulness of the proposed approach. Authors should consider
2. The procedure of selecting supporting items is quite heuristic, which is fine. But this also limits the novelty of the approach.

**Justification For Why Not Higher Score:**

The proposed method is limited in its impact and its practical usefulness is also not clear.

**Justification For Why Not Lower Score:**

N/A

---

### Decision · Program_Chairs · 2024-01-16

Reject